# The Role of Airborne Particles in the Epidemiology of Clade 2.3.4.4b H5N1 High Pathogenicity Avian Influenza Virus in Commercial Poultry Production Units

**DOI:** 10.3390/v15041002

**Published:** 2023-04-19

**Authors:** Joe James, Caroline J. Warren, Dilhani De Silva, Thomas Lewis, Katherine Grace, Scott M. Reid, Marco Falchieri, Ian H. Brown, Ashley C. Banyard

**Affiliations:** 1Department of Virology, Animal and Plant Health Agency (APHA-Weybridge), Woodham Lane, Addlestone KT15 3NB, UK; 2WOAH/FAO International Reference Laboratory for Avian Influenza, Animal and Plant Health Agency (APHA-Weybridge), Woodham Lane, Addlestone KT15 3NB, UK; 3Epidemiology and Risk Policy Advice, Advice Services, Animal and Plant Health Agency (APHA), Woodham Lane, Addlestone KT15 3NB, UK

**Keywords:** avian influenza virus, H5N1, avian, transmission, airborne, aerosol, droplet, dust

## Abstract

Since October 2021, Europe has experienced the largest avian influenza virus (AIV) epizootic, caused by clade 2.3.4.4b H5N1 high pathogenicity AIV (HPAIV), with over 284 poultry infected premises (IPs) and 2480 dead H5N1-positive wild birds detected in Great Britain alone. Many IPs have presented as geographical clusters, raising questions about the lateral spread between premises by airborne particles. Airborne transmission over short distances has been observed for some AIV strains. However, the risk of airborne spread of this strain remains to be elucidated. We conducted extensive sampling from IPs where clade 2.3.4.4b H5N1 HPAIVs were confirmed during the 2022/23 epizootic, each representing a major poultry species (ducks, turkeys, and chickens). A range of environmental samples were collected inside and outside houses, including deposited dust, feathers, and other potential fomites. Viral RNA (vRNA) and infectious viruses were detected in air samples collected from inside and outside but in close proximity to infected houses, with vRNA alone being detected at greater distances (≤10 m) outside. Some dust samples collected outside of the affected houses contained infectious viruses, while feathers from the affected houses, located up to 80 m away, only contained vRNA. Together, these data suggest that airborne particles harboring infectious HPAIV can be translocated short distances (<10 m) through the air, while macroscopic particles containing vRNA might travel further (≤80 m). Therefore, the potential for airborne transmission of clade 2.3.4.4b H5N1 HPAIV between premises is considered low. Other factors, including indirect contact with wild birds and the efficiency of biosecurity, represent greater importance in disease incursion.

## 1. Introduction

Since 2014, there have been repeated incursions of H5Nx HPAIV of the A/goose/Guangdong/1/96 (GsGd) lineage into the poultry sector across Europe, thought to be driven by the movement of infected wild bird species on migratory pathways. In Great Britain (GB), these incursions have been with multiple different subtypes and genotypes, primarily including H5N8 (2014/15, 2016/17, and 2020/21), H5N6 (2017), and H5N1 (2021/22/23) [1,2,3,4,5,6,7]. The 2021/22/23 epizootic began with the first poultry-infected premises (IP) in GB on 24 October 2021 [8], where it has continued to date, culminating in over 284 confirmed HPAIV H5N1 IPs and over 2480 wild birds found dead testing positive across GB [7,8]. These unprecedented infection levels have been mirrored across continental Europe [6,7] with the further spread of H5N1 HPAIV to the Americas [9,10].

Transmission of AIV may occur as a result of direct contact between infected and susceptible birds or through indirect contact via fomites [11,12,13]. Indirect contact or fomite transmission can be distinguished based on the matrix into solid, liquid, and airborne [11,14,15]. The most efficient route of transmission varies hugely depending on viral strain and viral survival characteristics, especially where environmental contamination is of concern [16]. In birds, transmission most frequently occurs via environmental contamination through liquid fomites (i.e., fecal–oral route) [12,13,17,18]. However, transmission via airborne particles may be relevant for some strains of AIV (e.g., H9N2) [13,14,19,20,21,22,23,24,25].

Airborne particles implicated in the transmission of AIV can be further sub-categorized based on their relative size and composition into (i) biological aerosols (≤5 μm) [19,20], (ii) biological droplets (>5 μm) [12,26], (iii) microscopic fomite particles (e.g., dust or animal dander) [20], or (iv) macroscopic fomite particles (e.g., feathers) [19]. Biological aerosols and droplets consist of microscopic biological fluids, such as those atomized from mucosal tracts (e.g., from the oropharyngeal [Op] or cloacal [Cl] tracts of birds) during normal behavior [12,27,28,29,30,31]. In contrast, fomite particles consist of biological or non-biological material (dust, dander, feather etc.) contaminated with a virus; these particles vary in size depending on their origin [32]. Depending on their composition and the environment, some airborne particles can be disseminated by wind and air currents, potentially over large distances, whereas others settle quickly over a smaller distance [12,15,33,34,35].

Multiple factors influence the efficiency of airborne transmission, and the hurdles to successful airborne transmission can be defined in three stages. Stage 1—Generation: airborne particles must be generated that harbor a sufficient titer of infectious virus [14,31,32,36,37,38,39]. Stage 2—Transport: airborne particles must be transported to a susceptible host, retaining a sufficient minimum infectious titer of the virus [33,34,35,40,41]. Stage 3—Infection: a sufficient titer of the virus must contact the ‘target’ cells of a susceptible host to initiate infection [13,42,43,44].

Airborne transmission over short distances between birds has been demonstrated for some AIV strains [13,14,19,20,21,22,23,24,25], and AIVs have also been detected in the air around wild birds in their natural habitat [14]. However, in the context of the GsGD lineage of viruses, H5Nx HPAIV airborne transmission among avian species has not been conclusively demonstrated. Limited airborne transmission of GsGD lineage H5 HPAIVs has been shown to occur experimentally over short distances [45,46]. In the field, H5Nx HPAIV viral RNA (vRNA) has been detected in air samples taken in the vicinity of live bird markets [47,48], and both vRNA and infectious H5 HPAIV have been found in air sampled inside of affected houses on IPs [49]. However, all these studies were with historic GsGD-lineage H5 HPAIV, distinct from the currently circulating clade 2.3.4.4b H5N1 HPAIV.

These reports, coupled with the observations that airborne particles are high in poultry houses and significant levels of particulate matter enter poultry houses through ventilation systems [40], raise the possibility of airborne spread of contemporary H5N1 HPAIV. Indeed, in GB, many of the reported IPs appear to cluster geographically and have occurred despite traditional biosecurity countermeasures (disinfectant ‘foot-dips’ and changes of outside clothing etc.) [7]. We previously demonstrated limited airborne detection of vRNA from ducks and chickens infected with contemporary 2021 H5N1 HPAIV clade 2.3.4.4b, albeit under experimental conditions [18]. However, currently, no data exists for the field detection of H5N1 HPAIV in airborne particles from the current global panzootic.

Here we investigated potential routes of onward transmission of clade 2.3.4.4b H5N1 HPAIV from infected poultry premises. We collected a range of environmental samples, including air, dust physically trapped on the ventilation systems, water, and feathers from three commercially infected premises of ducks (fattening ducks), turkeys (135-day-old meat turkeys), and chickens (38-day-old broilers), representing the three main commercially farmed poultry species globally, during the 2022 and 2023 clade 2.3.4.4b H5N1 HPAIV epizootic. Using accredited molecular and virological methods, we confirmed or negated the presence of vRNA and infectious HPAIV in samples collected, allowing us to infer the risk of transmission of H5N1 HPAIV between poultry premises.

## 2. Materials and Methods

### 2.1. Description, Clinical Picture, and Atmospheric Conditions of the Duck Infected Premises (IP1)

The first infected premises was in Norfolk, UK (Figure 1) and consisted of approximately 26,000 fattening ducks contained within two houses (1 and 2). The houses were identical; both houses were naturally ventilated, with openings underneath the overhanging roof down the entire east and west sides, covered by a metal mesh. There was a single large entrance to the south of both houses and standard door entry for personnel at the south and north of each house. There was a 0.5 m opening running down the length of the roof apex on both houses covered with metal mesh.

A non-negative H5 PCR pre-movement test result (to determine freedom from influenza in the birds for licensing purposes) was obtained from house 1; samples were collected by the private veterinary surgeon (PVS) on 13 December 2022 and officially confirmed on 16 December 2022. At the time of pre-movement sampling, the birds were reported to be clinically normal. On the afternoon of 16 December 2022, six ducks were found dead in house 1, and clinical signs, including lethargy and depression, were reported. Slaughter on suspicion (SoS) was declared on 16 December 2022. Culling of birds in house 1 began on 19 December 2022 at approximately 10:00 a.m. Clinical samples (Op and Cl swabs and brain samples) were collected for statutory diagnosis of notifiable avian disease (NAD) and returned positive detections of clade 2.3.4.4b H5N1 HPAIV. Environmental sampling began at approximately 12:15 p.m. on 19 December 2022. Environmental samples were taken to the laboratory and stored at 4 °C overnight prior to testing.

At the time of environmental sampling, there was heavy cloud coupled with some occasional light rain, and the ground was wet. There was a south south–west wind at 9 m/s, with gusts of 15 m/s [50]. The air temperature was 12 °C, and the UV index was low [51].

### 2.2. Description, Clinical Picture, and Atmospheric Conditions of the Turkey Infected Premises (IP2)

The second infected premises was in Norfolk, UK (Figure 2), consisting of approximately 20,500 135-day-old turkeys within three houses (1, 2, and 3) (Figure 2). All other houses were empty at the time of sampling. The houses were identical and mechanically ventilated, with four extract points down the north–west and south–east sides; there were six inlet vents on the roof apex of each house. There were single large entrances at the north–east sides of all houses along with a standard door entry for personnel.

On 8 January 2023, 45 dead turkeys were reported in house 1; no other clinical signs were apparent in the rest of the flock. The owner reported suspicion of NAD on 8 January 2023. On 9 January 2023 a further 105 turkeys were found dead in house 1 (total flock size of 6308 turkeys). Clinical signs in house 1 included respiratory and nervous signs, green diarrhea, lethargy, recumbency, and a significant decrease in water consumption. In house 3, mortality had also increased. No clinical signs were observed in turkeys in house 2 at the time of the initial investigation. Clinical samples (OP and C swabs and brain samples) were collected for statutory diagnosis of NAD and returned positive detections of clade 2.3.4.4b H5N1 HPAIV. Culling of house 1 began at approximately 09:00 on 12 January 2023. Environmental sampling began at approximately 11:30 a.m. on 12 January 2023 while the culling was underway, and there was high mortality in house 3 and low mortality in house 2.

At the time of environmental sampling, there was heavy cloud coupled with some occasional light rain, and the ground was wet with some standing ground water. There was a south-west wind at 6.7 m/s, with gusts of 13.4 m/s [50]. The air temperature was 12 °C, and the UV index was low [47].

### 2.3. Description, Clinical Picture, and Atmospheric Conditions of the Chicken Infected Premises (IP3)

The third infected premises was in Norfolk, UK (Figure 3). IP3 consisted of approximately 300,000 38-da- old broiler chickens within eight houses (1–8; approximately 37,500 per house). The houses were identical; each house was mechanically ventilated, with 16 extract points down the roof apex and 15 inlet vents on the north and south sides of the houses. There was a single large entrance at the east and west sides of all houses, along with a standard door entry for personnel.

On 19 February 2023, 265 dead chickens were reported in house 5; no other clinical signs were apparent in the rest of the flock. The owner reported suspicion of avian notifiable disease on 19 February 2023. On 20 February 2023, further 705 chickens were found dead in house 5. Reduced feed and water consumption were reported in houses 1 and 5. Birds in houses 1 and 5 were reported as being quiet and reluctant to move. Clinical Samples (OP and C swabs and brain samples) were collected for statutory diagnosis of NAD and returned positive detections of clade 2.3.4.4b H5N1 HPAIV. Environmental sampling began at approximately 10:30 a.m. on 21 February 2023 when there was high mortality in house 5 and low mortality in house 1.

At the time of environmental sampling, there was a light cloud. with a west–southwest wind at 3.6 m/s, and gusts of 7.2 m/s [50]. The air temperature was 9 °C, and the UV index was low–moderate [47].

### 2.4. Clinical Sample Collection and Processing

Swabs were used to collect samples from the Op and Cl cavity. Swabs were individually cut and expressed into 1 mL of Leibovitz’s L-15 Medium (Gibco; (LM) [52]), and the supernatant was used for RNA extraction or stored at −80 °C until further use.

### 2.5. Environmental Sample Collection and Processing

Sample collection was carried out in strict order from areas of low potential contamination to areas of high contamination in order to avoid biological contamination of samples. A total of eight air samplers were deployed strategically to collect air from locations whereby natural air flow might drive viral material out of the affected houses. The location of air sampling was based on the wind direction at the time of sampling. Samplers were set up according to the perceived likelihood of collecting viral material, with both air samples and environmental samples being set up outside the houses before entering the areas of higher risk for viral exposure. The air samplers consisted of Gelatin Filters (25 mm, with a 3 µm pore size) housed in a Button Aerosol Sampler (SKC Ltd.), with each connected to an APEX2 air pump (Casella). Air was drawn for varying durations at a flow rate of 5 L/min, and the total volume of air sampled was recorded. While the filter has a nominal pore size of 3.0 μm, these filters have a higher capture efficiency of sub-micron particles through inertial impaction and diffusional interception [53]. This enables the immobilization of inhalable particles (size range: <1–100 μm) which include a range of bioaerosols and dust. The gelatin filters were removed aseptically into a sterile falcon tube and transported to the laboratory, where they were held at 4 °C overnight. The filters were dissolved by vortexing in 2 mL Leibovitz’s L-15 Medium (Gibco; (LM) [52]), and total RNA was extracted from the supernatant.

For dust and other solid material, approximately 1 g of material was added to 1 mL of sterile phosphate-buffered saline (PBS) (Sigma) and vortexed. With the feather samples, 1 mL of PBS was added to an individual or pooled feather sample and vortexed. RNA was extracted from the supernatants obtained from the feather and solid matrices and directly from the water samples [54]. For dust and other solid material, approximately 1 g of material was added to 1 mL of PBS and vortexed. For feather samples, 1 mL of PBS was added to individual or pooled feather samples and vortexed. RNA was extracted from the supernatants obtained from the feather and solid matrices and directly from the water samples [54].

### 2.6. RNA Extraction and AIV Reverse Transcription Real-Time PCR (RRT-PCR)

RNA was extracted from the environmental samples using the MagMAXTM CORE Nucleic Acid Purification Kit (ThermoFisher Scientific™) as part of the robotic Mechanical Lysis Module (KingFisher Flex system; Life Technologies), according to the manufacturer’s instructions. 2 µL volumes of extracted RNA were tested by the H5 HPAIV detection RRT-PCR using the primers and probes designed by James, Seekings [50]. RRT-PCR Ct values < 36.00 were considered as AIV positive, and sub-threshold values in the range Ct 36.01–39.99 and Ct 40.00 (“No Ct”) were interpreted as negative. A ten-fold dilution series of titrated H5N1-21/22 HPAIV RNA was used to construct a standard curve using Agilent AriaMx software (Agilent, UK) to determine PCR efficiency, which assured optimal assay performance for quantitative interpretation.

### 2.7. Virus Isolation and Propagation

For each sample, 100 µL of material was added to 100 µL of PBS containing a mixture of antibiotics (gentamycin, 50 mg/L; penicillin G, 1 million units/liter; streptomycin sulphate, 10 g/L; nystatin, 5 million units/liter [Sigma]). The sample was incubated for 1 h, and 100 µL was inoculated into the allantoic cavity of two specific pathogen-free (SPF) 9-day-old embryonated fowls’ eggs (EFE), as described previously [55]. At 2 days post-inoculation (dpi), the allantoic fluid of one EFE was collected and tested for the presence of a hemagglutinating agent using the hemagglutinin assay (HA) as previously described [55]. If no HA activity was observed at 2 dpi, allantoic fluid from the remaining EFE was collected at 6 dpi and again tested by HA. HA activity >1/4 at either 2 or 6 dpi was considered positive for virus isolation. Conversely, HA activity ≤1/4 at both 2 and 6 dpi was considered negative for virus isolation.

## 3. Results

### 3.1. H5N1 HPAIV Detection in Environmental Samples Collected at the Infected Duck Premises (IP1)

Four air samples were collected within 1 m of the outside of the vents (samples 3–6) to the north and north–east sides of house 1 and house 2, respectively (Figure 1). One air sample was collected approximately 10 m to the north–east of the IP, behind house 2 (Figure 1), and one air sample was collected approximately 120 m north–east from the back of the IP (Figure 1). In total, two separate air samples were collected from inside houses 1 and 2. During collection, elevated levels of mortality were seen across ducks in house 1, and culling had almost completed during collection. Air was collected from house 1 during culling at a time when birds were being collected and transported for a controlled atmosphere stunning in units outside of the houses but on the premises (Figure 1). All but two of the air samples tested positive for vRNA; one taken to the north of house 2 and one sample collected approximately 120 m from the houses was negative for vRNA (Appendix A and Figure 1). Infectious virus was isolated from both air samples collected from inside both houses (samples 1 and 2; Appendix A and Figure 1). The only other air sample where the infectious virus was detected was outside, within 1 m to the north-east of house 1 (Appendix A and Figure 1). All other air samples, including those collected at approximately 10 m or 120 m from the IP, were negative for infectious virus (samples 7 and 8; Appendix A and Figure 1).

Dust was collected from both outside and inside the houses. Particles of dust had visibly deposited on the mesh surrounding the vents all along the east and west sides of both houses. In total, twelve dust samples were collected by physically removing dust which had collected on the mesh of both houses (3 per side) (Figure 1). The samples consisted of fine particulate matter that is often trapped in cobwebs along the mesh. All but one of the dust samples collected from outside the vents tested positive for vRNA; one dust sample collected from the west of house 2 tested negative for the presence of vRNA (Appendix A and Figure 1). Infectious virus was detected from one out of six dust samples collected from the vents of houses 1 and 2 (Appendix A and Figure 1). Only the dust sample collected from the south–east side of house 2 was positive for the infectious virus.

In total, two water samples were collected from standing water that had accumulated on the concrete following recent rainfall. Both were collected outside to the south entrance of house 1 and house 2, where vehicular traffic was passing as part of the culling activities (Figure 1). The two water samples and two wet straw samples tested negative for vRNA, and virus isolation was not performed (Appendix A and Figure 1).

Two mature feathers, matching the morphology of duck feathers, and measuring approximately 5 cm in length, were collected and pooled, both located outside to the east of house 2 within 60 m from the cull site. These feathers were representative of those which were observed being carried on the wind following culling activities. The pooled feathers tested positive for vRNA but negative for the infectious virus (Appendix A and Figure 1).

One Op swab from a dead duck in house 2 was collected to serve as a positive control which tested positive for vRNA and infectious virus (sample 26; Appendix A and Figure 1).

### 3.2. H5N1 HPAIV Detection in Environmental Samples Collected at the Infected Turkey Premises (IP2)

In total, three air samples were collected down-wind of the affected houses and culling location; one air sample was approximately 120 m away (sample 8) and two approximately 50 m away, either down-wind of the affected houses or the culling location (sample 6 and 7, respectively) (Figure 2). vRNA was not detected in any of these samples. Three air samples were collected within 1 m of the outside of the houses (samples 3–5); two were collected to the north–east side (front) of houses 1 and 3, respectively; one was collected from under the vent extract to the south of house 3 (sample 5) (Figure 2). vRNA and infectious virus were only detected from the extract air vents of house 3 (Figure 2, Appendix A). An air sample was collected from within houses 2 and 3 (samples 1 and 2), both samples were positive for vRNA, but only the sample from house 3 also tested positive for the infectious virus (Figure 2 and Appendix A).

Dust was collected from outside and inside all three houses. The dust had visibly deposited on the ground and surfaces surrounding the outside and inside of the vents at four extract points along the south–east and north–west sides of the three houses (Figure 2). In total, eight dust samples were collected from the outside extracts by physically removing dust which had collected (samples 22, 23, 25, 26, 29, 30, 31, and 34) (Figure 2 and Appendix A). The samples consisted of fine, dry particulate matter. Four out of the eight external dust samples tested positive for vRNA, with at least one sample per house testing positive, although no infectious virus was detected in any sample. Dust samples were also collected from the inside surfaces (samples 21, 28, and 33) and from the internal extract vents (samples 24, 27, and 32) (Figure 2 and Appendix A); all but one sample tested positive for vRNA; sample 28, taken from inside of house 2 tested negative. No infectious virus was detected in any dust samples collected from internal surfaces.

Feathers were seen being carried by the wind following culling and carcasses being loaded into lorries for disposal, despite feather dampening being observed. Twelve feathers, matching the morphologies of turkey feathers, and being representative of the dispersed feathers, were collected at varying locations away from the cull site from 60 m to 180 m (Figure 2 and Appendix A). Feathers were selected based on having the same morphology as turkey feathers. Out of the 12 feathers, 3 tested positive for vRNA, with the maximum distance of a positive feather being 80 m from the cull location; no infectious virus was detected.

In total, six water samples were collected from the IP. Five from standing water, following recent rainfall, outside the houses (sample 35 to 38) and one from the drinking water from within house 3 (sample 39) (Figure 2 and Appendix A). One of the outside water samples, collected outside and under the vents of house 3, tested positive for vRNA but not for the infectious virus (sample 40) (Figure 2 and Appendix A). The drinking water inside house 3 (sample 39) also tested positive for vRNA and an infectious virus.

Op swabs were collected from dead turkeys in houses 2 and 3 to serve as a positive control; both samples tested positive for vRNA and an infectious virus (Appendix A).

### 3.3. H5N1 HPAIV Detection in Environmental Samples Collected at the Infected Chicken Premises (IP3)

Three air samples were collected down-wind of the affected houses, being taken approximately 70 m, 25 m and 5 m away from house 5 (samples 8, 7, and 6, respectively) (Figure 3). vRNA was not detected in any of these samples. Two air samples were collected on the roofs and down-wind of the extract vents from houses 1 and 5 (samples 3 and 4, respectively); one sample was collected from under an open vent to the north of house 5 (sample 5) (Figure 3), again no vRNA was detected in any of these samples. Air samples were collected from within houses 1 and 5, with only the air sample collected from within house 5 (sample 2) being positive for vRNA but not an infectious virus (Figure 3 and Appendix A).

Dust was collected from outside and inside all three houses. The dust had visibly deposited on the surfaces surrounding the outside and inside of the vents at 16 openings along the north and south sides of the two houses (Figure 3). A total of 18 dust samples were collected from the outside extracts in total by physically removing dust which had collected (samples 16–33) (Figure 3 and Appendix A). The samples consisted of fine, dry particulate matter. Only one sample (sample 26) tested positive for vRNA, but not an infectious virus. Dust samples were also collected from the inside surfaces of these openings (samples 34–35, 38–40) and from the surfaces inside the houses (samples 37 and 41–42) (Figure 3 and Appendix A); none of these samples tested positive for vRNA.

Feathers, with similar morphologies to chicken feathers, were found outside the houses on the central road and down-wind of house 5 (Figure 3). Five feathers, representative of the dispersed feathers, were collected at varying locations (Figure 2 and Appendix A). One of these feathers outside the houses tested positive for vRNA, located close to the outside of house 5 (sample 12). Feathers were also collected from the floor inside houses 1 and 5, although only the sample from house 5 tested positive for vRNA (sample 10). No infectious virus was detected in any feather sample.

In total, two water samples were collected from within houses 1 and 5, both from the drinking water receptacles. Only the sample from within house 5 tested positive for vRNA and infectious virus (Appendix A). No other environmental water sources were apparent on the IP.

Op and Cl swabs were collected from dead chickens in houses 1 and 5 to serve as a positive control. Both swabs from house 5 tested positive for vRNA and infectious virus (Appendix A).

## 4. Discussion

Limited studies have described the potential risk of dissemination of H5 HPAIV within the environment. No accounts of field airborne detection or transmission among birds with the contemporary HPAIV H5N1, representative of the 2021 to 2023 epizootic, have been reported. To provide greater insight into these potential transmission routes, we collected and tested over 115 environmental samples across three premises confirmed to be infected with H5N1 HPAIV during the 2021 to 2023 epizootic in GB. These premises were representative of commercial duck, turkey, and chicken premises—the three most significant poultry species globally, and consisted of a range of naturally and mechanically ventilated configurations. We detected HPAIV RNA in air and in dust samples outside of affected houses across all IPs, up to a maximum distance of 10 m, with infectious virus only being detected sporadically, up to only 1 m from infected sheds.

A few historical studies have detailed the detection of H5Nx HPAIV RNA in the air in and around IPs [13,47,48,49]. However, these studies have assessed airborne detection of comparatively historic viruses with different transmission efficiencies. In addition, many of these studies have either not assessed or have failed to detect an infectious virus. In experimental studies, historic H5Nx HPAIVs have been reported to be able to transmit between birds physically separated yet sharing the same airspace over relatively short distances (up to 1.1 m) during normal behavior [45] or following butchery practices [56]. A more recent field study by Filaire, Lebre [49] detailed the detection of vRNA present in surface dust and air samples inside houses across 63 premises, including 30 officially declared IPs, 17 officially negative premises and 16 unconfirmed premises [49]. All premises were geographically linked or had epidemiologic links to known IPs infected with 2020/21 Eurasian H5N8 HPAIV. Each of these premises was selected based on the severity of clinical signs exhibited (ranging from absent to severe). A small subset of samples tested positive for the presence of infectious viruses in either air or dust samples [49]. While the chronology of environmental sampling and infection of the flock is difficult to interpret, Filaire, Lebre [49] did detect vRNA in dust or air samples from two premises officially declared negative by the competent authority in France, suggesting the presence of vRNA without active infection in the flock. This potentially indicates airborne spread from adjacent or neighboring IPs, but other factors, including mechanical transfer by human activity or introduction from a wild bird origin, cannot be discounted. However, Filaire, Lebre [49] did not collect samples outside of affected poultry houses; therefore, the distance virus could potentially travel from an affected house was not assessed.

In the current study, H5Nx HPAIV vRNA was detected in the air samples taken within houses containing infected birds across all three IPs, but the infectious virus was only isolated from air sampled at the duck and turkey premises. Air samples taken outside but adjacent to, or down-wind of, affected houses varied in the presence of H5Nx vRNA, with the duck premises showing higher positivity, 75% (3/4) of air samples collected within 1 m of the boundary of the duck houses contained vRNA, and 25% (1/4) contained an infectious virus. Air samples taken from similar locations at the turkey and chicken premises revealed 33% (1/3) and 0% (0/3) vRNA positivity, respectively, with none of these samples containing an infectious virus. These findings suggest that microscopic airborne particles harboring viable infectious H5N1 HPAIV can only travel short distances outside of a poultry house in detectable levels, ≥1 m but <10 m. However, either through inactivation prior to becoming airborne or in the air, microscopic airborne particles containing viral RNA may travel further, up to distances of ≥10 m but <25 m.

Several studies have investigated the distance the H5 HPAIV virus can travel through the air in a field environment. HPAIV RNA has been detected in air samples taken in the vicinity of live bird markets, several meters from the birds [47,48]. In commercial poultry premises, sampling revealed the presence of H5N2 HPAIV RNA detected at 1000 m from an infected house, but the infectious virus was only detected from samples collected within the house and up to 70 m away [47]. Virus RNA has also been detected in samples taken up to 110 m from a duck IP and inside infected poultry houses during the more closely related 2016 H5N8 epizootics [48]. However, these viruses are all distantly related to the current clade 2.3.4.4b H5N1 HPAIV from the 2021 to 2023 epizootic, and therefore transmission dynamics via the air may also differ, dependent on factors such as the total virus being released into the environment from birds which have been observed to differ for even more closely related viruses in experimental studies [18,52,54,56]. Alongside HPAIVs, other reports have documented the detection of low levels of low pathogenicity AIV (LPAIV) RNA 60 m down-wind from infected premises, but an infectious virus was not detected [38]. Importantly, in the study presented here, only one sample taken at 10 m or greater tested positive for vRNA; this sample was taken 10 m down-wind of an affected duck house; the next furthest air sample, taken at 120 m, was negative for vRNA. All other air samples collected at greater than 10 m tested negative for vRNA and infectious virus across all three premises. The air sampling results closely mirrored the dust samples, where HPAIV RNA was detected in at least one dust sample deposited outside but close to the ventilation systems of the houses across all three IPs. Infectious virus was only detected in the samples collected from the duck premises. We have previously reported that vRNA could not be detected in air samples collected following experimental infection of ducks and chickens with contemporary H5N1 2021 HPAIV clade 2.3.4.4b [18]. However, these animals were housed under negative air pressure within a high containment environment which may have impacted airflow and the ability to detect particulate material bound to infectious viruses or viral products.

For a successful airborne transmission to occur, multiple hurdles must be overcome, defined broadly as airborne particles: (i) generation, (ii) transport, and (iii) infection. Current clade 2.3.4.4b H5N1 HPAIV can reach high titers in both Op and Cl cavities, secretions from which have been demonstrated to contaminate the immediate housing environment, including bedding material, at titers exceeding a proposed minimum infectious dose [18]. Moreover, poultry operations are known to produce high concentrations of dust generated from bedding, food, dander, feather material and fecal matter [37,57]. Consequently, the likely source of the detection of vRNA and infectious viruses in environmental samples in this study is due to these contaminated sources being aerosolized. Interestingly, we did observe greater detection of HPAIV in the air from the duck, compared to the turkey and chicken premises. This observation mirrors experimental findings where ducks have exhibited particularly elevated levels of environmental contamination following experimental infection with AIVs in comparison to other poultry species such as Galliformes (chickens and turkeys) [18,54,58].

There were no significant differences in the positive Ct values (and extrapolated vRNA quantity) obtained from inside or around the houses, with similar volumes of air being sampled at equivalent locations for both houses, indicating that culling activities may not necessarily enhance the likelihood of H5N1 HPAIV dissemination in the air. Indeed, the air sample taken from IP1 (duck premises) house 2, where culling had not commenced, was positive to a similar level as samples taken during culling activities in house 1. Therefore, human activities which may be considered to aerosolize particles, such as culling, husbandry or bird transport, do not appear to increase the detectable concentration of virus or vRNA in the air. While we assayed for the presence of an infectious virus, we did not quantify the infectious virus in these samples; thus, the infectious titer was not determined. However, with reference to previous comparison data between infectious virus and vRNA concentration (relative equivalency units [REUs]), we estimate that the lowest Ct values (highest vRNA quantity) collected from the air samples may be below the equivalent minimum infectious dose for ducks [18]. Although these assumptions are based on the minimum infectious dose following intranasal inoculation and not from aerosols, the latter has been shown for historic H5N1 HPAIVs to be around 30 times less efficient [39]. Thus, the exact dose which would be received by poultry, although the air remains undefined, as does the minimum infectious dose via airborne routes of infection with contemporary clade 2.3.4.4b H5 HPAIV. Whilst volumes collected in the air varied across locations, all air volumes sampled are far in excess of the respiratory volume of common domestic poultry [59], and so the levels of RNA and infectious virus detected would be far greater than the levels inhaled by individual birds [59]. In addition, the detection of an infectious virus in the air inside the houses instills the importance of appropriate respiratory protection for staff entering these areas once NAD has been confirmed to prevent potential zoonotic infection.

Environmental conditions, such as the relative humidity, temperature, ultra-violet (UV) radiation (sunlight), particle composition, and chemical composition of the air can affect the infectivity of airborne viruses, and each virus reacts in its own way to each factor or combination of factors [60,61,62]. Several studies have demonstrated the half-life (time taken to reduce the viral titer by 50%) of influenza viruses to be between 2.4 to 31.6 min in airborne particles depending on particle size and environmental conditions [33,37,63,64]. However, there is currently no data available for virus survival in matrices found in the air of poultry premises. In this study, environmental conditions were similar at the time of sampling across all premises. The detection of an infectious virus did not necessarily correlate with the vRNA titer (Ct value), e.g., with the physical dust samples. This is likely because of differences in virus inactivation and survival in these sample types. We also observed a trend for fewer infectious virus-positive samples as the distance from the premises increased; viral RNA followed a similar trend yet at slightly further distances. We hypothesize this is due to inactivation as the virus travels through the air, but inactivation may also occur prior to aerosolization. It is also likely that virions are less protected from inactivation in smaller particles, which may travel further than in larger ones due to greater exposure to environmental conditions. In this study, particle size and type were not investigated, but future investigations into which particle sizes harbor viruses, as well as virus survival kinetics in different sample types, may provide a greater understanding of the epidemiological significance of this potential role of transmission. Consequently, uncertainty remains as to whether there is enough virus present in the air required to initiate a new infection. It is also important to note that these analyses were conducted with similar environmental conditions at each IP. On the other hand, these conditions as largely representative of those experienced in winter of temperate countries, different climactic conditions experienced in other regions or at different times of the year will likely affect the propensity of transmission via this route.

While a range of environmental and virological factors determine the survival and correlation between infectious viruses and vRNA, this can also be influenced by the sampling method. In this study, we used gelatin filters with a 3µm pore size [53]. This method is well established to capture virus-laden particles in the air, including less than 3µm because of biophysical interactions at the filter interface, and includes the detection of inhalable and respirable dust as well as biological aerosols for a wide range of respiratory pathogens, while best-preserving virus infectivity [18,53,65,66]. Moreover, the positive detection of vRNA and the infectious virus within the houses highlights the utility of this method in terms of virus recovery. However, each sampling approach carries its own limitations [67]. Despite this, similar conclusions around risk can be drawn from this study regardless of if only the infectious virus and vRNA or only the vRNA detection is used.

The dispersion of particles from infected premises will be influenced by many factors, including the ventilation rate, particle settling velocity, effective release height, wind speed and wind turbulence [68]. However, while ventilation rates have been shown to affect the trajectory and distance of particulate matter following release, they do not necessarily correlate with differences in total emissions [69]. It has been hypothesized that high rates of ventilation may lead to more frequent emissions, yet with a lower concentration of particles being released at any one time [38,69]. In this study, we performed an analysis of naturally ventilated premises (IP1, ducks) and two mechanically ventilated configurations (IP2, turkeys and IP3, and chickens), while the naturally ventilated configuration yielded greater positivity outside than the two which were mechanically ventilated, many other confounding variables exist, and variations in particle release via different ventilation requires further study as does the impact of particle size and type on vRNA or live virus dissemination.

Alongside microscopic fomites, we also investigated the potential transmission via larger, ‘macroscopic’ airborne fomites, including feathers. HPAIV, including the contemporary H5N1, has been demonstrated to replicate and reach high viral titers in feathers [18,54,70,71]. Moreover, HPAIV has been shown to survive in detached feathers for up to 15 days in ambient conditions [26]. Furthermore, because of the structure and capability for virus ingress at contact points on the bird, feathers may be expected to contain a higher concentration of virus per ‘particle’ than other airborne particles [72]. Consequently, feathers have long been regarded as potential sources of HPAIV infection and an elevated risk for infection transmission [26,36]. In this study, we detected vRNA in feathers across all IPs, up to 80 m from affected houses, although no infectious virus was detected. However, we only assessed the surface contamination present on these samples rather than the virus potentially sequestered in the internal calami. The association of viruses within feathers may shield the virus from environmental-inactivating factors that might reduce infectivity in other environmental studies [73]. Therefore, the risk of this route in potential bioavailability for the onward spread of infection may be underrepresented in the current study, and more work is required to investigate this route.

Together, these data support the hypothesis that infectious H5N1 HPAIV can be carried short distances (<10 m) through the air, whilst airborne particles harboring vRNA may travel further at detectable levels (≥10 m but <50 m [≥80 m but <100 m for feathers]). This suggests that the virus may be transmitted between houses and hence gives a further mechanism for sequential infection of houses at premises and is supported by infectious virus being detected within air samples taken within 1 m of the houses’ outer perimeter. It also raises the possibility that human behavior outside but close to affected houses could contribute to the transmission through the transfer of infectious material in fomites (i.e., through the mechanical transfer of external dust on clothing or equipment). Overall, the contribution of airborne transmission in the epidemiology of contemporary clade 2.3.4.4b H5N1 HPAIV from one IP to another poultry premises is considered to be very low, and other factors, such as the high incidence of infection in wild birds, environmental contamination by wild birds and poultry following the establishment of an infection within premises, and human behaviors in the time period where the virus has been introduced but the clinical disease has not yet been detected may all be of greater importance. Further studies are warranted across a broad range of husbandry systems to include the assessment of virus and viral product dissemination. Ideally, the different sectors and species require an assessment to improve our understanding of the potential for translocation of an infectious virus through either natural environmental distribution or via artificial mechanisms linked to human behaviors to have a more thorough evaluation of the risk posed by this potential transmission route.

## Figures and Tables

**Figure 1 viruses-15-01002-f001:**
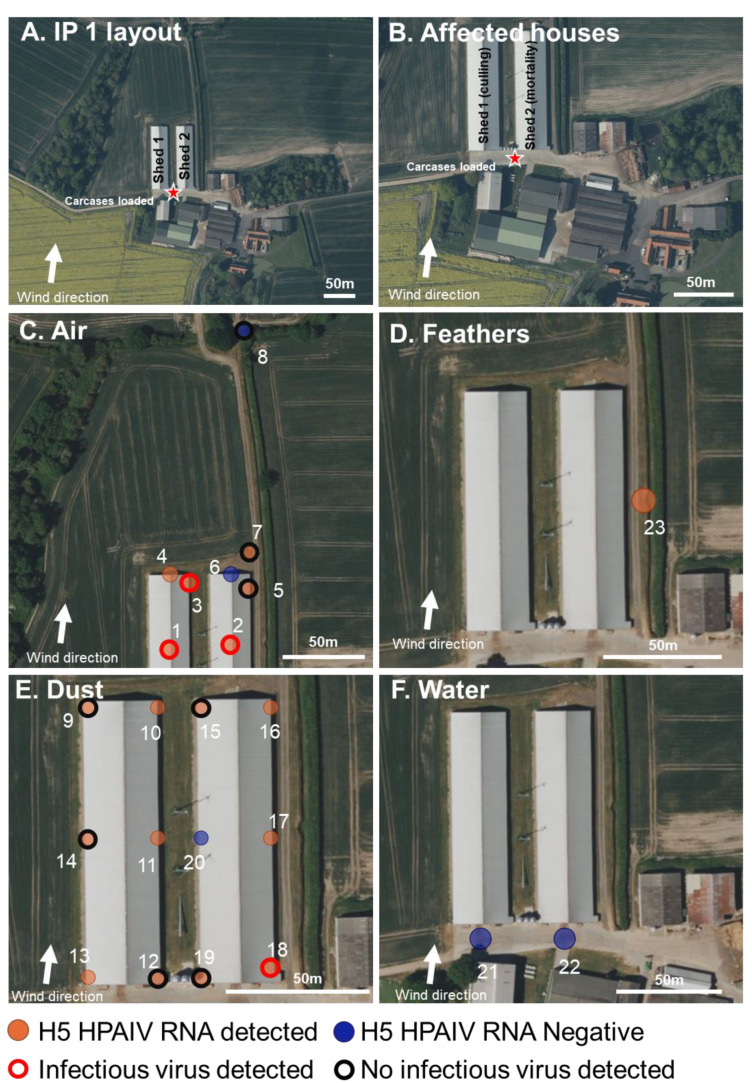
Infected premises 1 (IP1): Environmental sampling and presence or absence of H5 HPAIV RNA at a duck IP. (**A**,**B**). Map showing the layout of the IP with the two affected houses indicated, wind direction at time of sampling, the site where carcasses were loaded onto trucks (red star), and mortality and culling indicated. (**C**–**F**). The location of the environmental sampling and H5 RRT-PCR results showing the presence (orange) or absence (blue) of H5 HPAIV RNA in the sample and the positive isolation of infectious virus (red outline) or negative isolation of infectious virus (black outline). Samples were collected from air filters (**C**), feathers (**D**), dust (**E**), and water (**F**). Locations of wet straw samples (n = 2) and an Op swab from a dead duck (n = 1) are not shown but are indicated in Appendix A.

**Figure 2 viruses-15-01002-f002:**
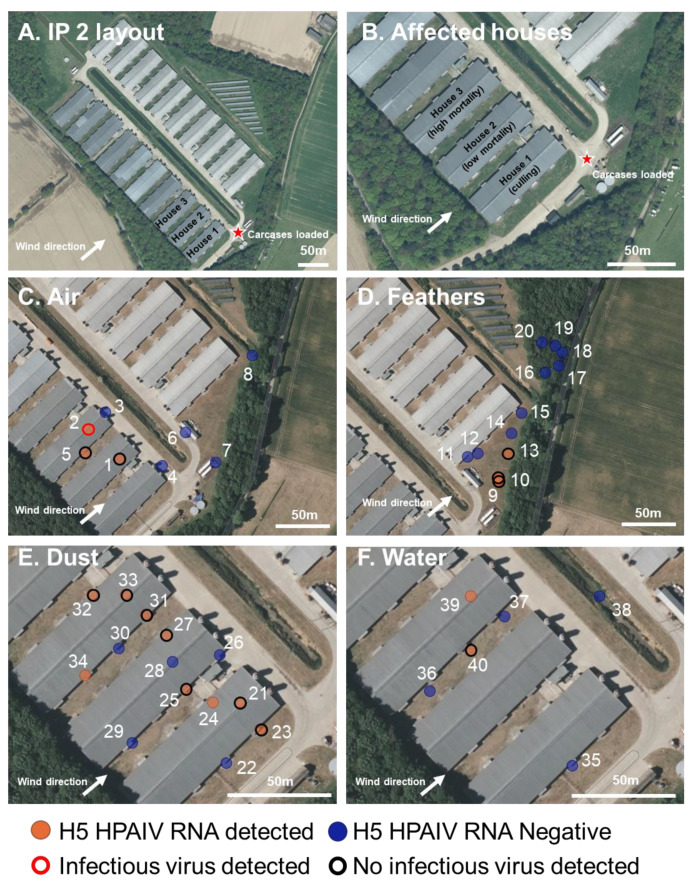
Infected premises 2 (IP2): Environmental sampling and presence or absence of H5 HPAIV RNA at a turkey IP. (**A**,**B**). Map showing the layout of the IP with the three affected houses indicated, wind direction at time of sampling, the site where carcasses were loaded onto trucks (red star), and mortality and culling indicated. (**C**–**F**). The location of the environmental sampling and H5 RRT-PCR results showing the presence (orange) or absence (blue) of H5 HPAIV RNA in the sample and the positive isolation of infectious virus (red outline) or negative isolation of infectious virus (black outline). Samples were collected from air filters (**C**), feathers (**D**), dust (**E**), and water (**F**). Op swabs from dead turkeys are not shown but indicated in Appendix A.

**Figure 3 viruses-15-01002-f003:**
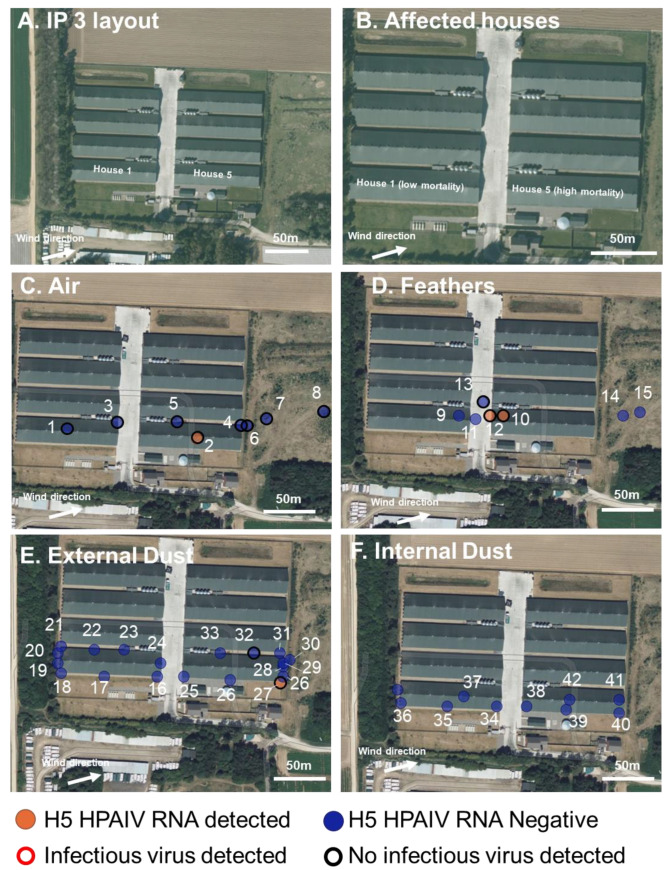
Infected premises 3 (IP3): Environmental sampling and presence or absence of H5 HPAIV RNA at a turkey IP. (**A**,**B**). Map showing the layout of the IP with the three affected houses indicated, wind direction at time of sampling, and mortality indicated. (**C**–**F**). The location of the environmental sampling and H5 RRT-PCR results showing the presence (orange) or absence (blue) of H5 HPAIV RNA in the sample and the positive isolation of infectious virus (red outline) or negative isolation of infectious virus (black outline). Samples were collected from air filters (**C**), feathers (**D**), external dust (**E**), and internal dust (**F**). Water and Op and Cl swabs from dead chickens are not shown but indicated in Appendix A.

## Data Availability

Not applicable.

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
