# Peer review of "The Role of Airborne Particles in the Epidemiology of Clade 2.3.4.4b H5N1 High Pathogenicity Avian Influenza Virus in Commercial Poultry Production Units"

_viruses, 2023, doi:10.3390/v15041002_

Round 1

Reviewer 1 Report

The manuscript by James et al., described the role of airborne transmission on H5N1 subtype AIV infection. In this manuscript, author find that the distance is the important factor for airborne transmission, and they draw a conclusion that the potential for airborne transmission of H5N1 HPAIV between premises is considered low. However, AIV epizootic caused by clade 2.3.4.4b H5N1 might be related the other factors, such as indirect contact with wild birds, the efficiency of biosecurity, etc. The data in the manuscript provide evidence for the possibility of influenza virus transmission in the air, and have certain reference value for the clinical avian influenza prevention and control.

However, there are several problems as follows:

1. Is the environmental sample representative?

2. Several formats in the article need to be modified, such as  and â„ƒ.

3. There is a grammar error in line 213 of the article.

4. The discussion part of the article needs to be reorganized. The human, geographical position and climate factors to cause the spread of the virus need to be discussed.

Author Response

We thank the reviewer for the time and comments to improve the manuscript. Please see below the replies to each comment.

  1. Is the environmental sample representative?

We have collected over 115 environmental samples across three different infected premises representing key poultry housing set ups in the UK. The sample types collected are based on previous sample types likely to harbor infectious virus, or be contaminated, following experimental infection (please see linked references below). While we agree that these samples are only a subset of all samples available on the IP, these samples have been collected in a methodical way, utilizing experience from experimental and field sampling efforts previously, therefore these samples are highly representative of those likely to harbor virus or be important in airborne transmission. We would like to also state that all these samples were collected and processed under high biological containment (BSL-3-Ag equivalent) levels using full respirator suits in the field which is technically challenging and intensive.

Clade 2.3.4.4b H5N1 high pathogenicity avian influenza virus (HPAIV) from the 2021/22 epizootic is highly duck adapted and poorly adapted to chickens | bioRxiv

Highly pathogenic avian influenza virus H5N6 (clade 2.3.4.4b) has a preferable host tropism for waterfowl reflected in its inefficient transmission to terrestrial poultry - PubMed (nih.gov)

Transmission dynamics between infected waterfowl and terrestrial poultry: Differences between the transmission and tropism of H5N8 highly pathogenic avian influenza virus (clade 2.3.4.4a) among ducks, chickens and turkeys - PubMed (nih.gov)

  1. Several formats in the article need to be modified, such as ≤and â„ƒ.

We thank the review for this comment, these have been reviewed and changed accordingly.

  1. There is a grammar error in line 213 of the article.

This has been changed

  1. The discussion part of the article needs to be reorganized. The human, geographical position and climate factors to cause the spread of the virus need to be discussed.

We thank the review for their comments and believe that these fairly minor amendments have strengthened our article. We have discussed these factors, including human behaviours in relation to culling activities (section from line 473), geographical and climate factors (section from line 497). However, we have added additional text to expand on of these factors in more detail.

Reviewer 2 Report

In this manuscript, James et al present data on the airborne transmission of highly pathogenic avian influenza H5 clade 2.3.4.4b viruses in natural settings. In brief, their studies demonstrate that infectious RNA can be found in the environment including water sources, feathers, and at short distances from the source. These are compelling and important studies that will have broad scientific and practical applications. The manuscript is well-written and the study design is rigorous. The differences in transmission between the houses and species is intriguing and requires further study. This is outside the scope of this work. 

Author Response

We thank the reviewer for their time and comments on the manuscript.

Reviewer 3 Report

Overall a good descriptive manuscript that studies the risk associated with airborne and environmental transmission of clade 2.3.4.4b H5N1 high pathogenicity avian influenza virus through use of aerosol and environmental samples. Requires some editing for language and formatting (see notes below). The study provides useful information in the study of avian influenza viruses, specifically relating to biosecurity practices and understanding how outbreaks can develop. In general, data here supports the idea that the virus will not remain viable for a long period of time in an aerosol or the environment, limiting the involvement of transmission through environmental contamination. The authors recognise where limitations exist and advise where further research could be carried out.   

Line 176 – bold formatting?

Line 181 – bold formatting?

Use of degree symbol with temperatures – line 174/154 and others throughout manuscript.

Use of dates – no need to say ‘the 08/01/23’ on date will suffice.

Line 182 – 183. Simpler to say sample collection was carried out in strict order from areas of low potential contamination to areas of high contamination, in order to avoid biological contamination of samples?

Line 213 – 2 two ul – can this be rephrased to make clearer? (tested in duplicate?)

Line 223 – list antibiotics (should be able to replicate study from methods)

Fig 1 – Red text is difficult to read – can colour be changed, or rely on annotation in legend?

First paragraphs of results sections 3.1, 3.2 and 3.3 could be summarised in tables – would also be easier for reader to follow and avoid repeating methods. Visual representation of this in fig 1, 2 and 3 is useful.

Line 390-397 Repetition from introduction/methods. Not necessary here?

Line 427- should be 75%?

Line 431-434. This implies that viral RNA and infectious virus are separate entities. Is this the case and if so, how has this been shown (fragments rather than whole genome?)?  These results suggest that the virus only remains viable for a short distance/period away from infected birds. Airborne particles containing virus may travel great distances (or only larger airborne particles contain virus, but there is no sizing differentiation in sampling in this study), but what happens to the virus is the pertinent thing – virus in airborne particles is inactivated/degraded beyond the point of recognition in detection assays. Could be clearer on this topic here.

Line 510 – should be gelatine?

Line 520 – should be ‘influenced by many factors’

Author Response

We thank the reviewer for their time and valuable comments to improve the manuscript please see the responses to the comments provided below.

Line 176 – bold formatting?

This has been changed

Line 181 – bold formatting?

This has been changed

Use of degree symbol with temperatures – line 174/154 and others throughout manuscript.

This has been changed

Use of dates – no need to say ‘the 08/01/23’ on date will suffice.

This has been changed

Line 182 – 183. Simpler to say sample collection was carried out in strict order from areas of low potential contamination to areas of high contamination, in order to avoid biological contamination of samples?

We thank the reviewer for their suggestion, we have amended the text accordingly

Line 213 – 2 two ul – can this be rephrased to make clearer? (tested in duplicate?)

This has been changed

Line 223 – list antibiotics (should be able to replicate study from methods)

We have added the list of antibiotics included

Fig 1 – Red text is difficult to read – can colour be changed, or rely on annotation in legend?

We thank the review for this suggestion, this has been changed to improve readability

First paragraphs of results sections 3.1, 3.2 and 3.3 could be summarised in tables – would also be easier for reader to follow and avoid repeating methods. Visual representation of this in fig 1, 2 and 3 is useful.

We thank the review for this comment, we feel this adds to the description of the results and provides context for the interpretation. We have summarised these data in supplementary figures, but would like to retain for completeness.

Line 390-397 Repetition from introduction/methods. Not necessary here?

We have removed a large section of text.

Line 427- should be 75%?

Thank you, this has been amended

Line 431-434. This implies that viral RNA and infectious virus are separate entities. Is this the case and if so, how has this been shown (fragments rather than whole genome?)?  These results suggest that the virus only remains viable for a short distance/period away from infected birds. Airborne particles containing virus may travel great distances (or only larger airborne particles contain virus, but there is no sizing differentiation in sampling in this study), but what happens to the virus is the pertinent thing – virus in airborne particles is inactivated/degraded beyond the point of recognition in detection assays. Could be clearer on this topic here.

Detection of viral RNA in the absence of infectious virus indicates that the virus which was present has been inactivated. We agree with the reviewer that it is difficult to delineate at which point this inactivation may have occurred, either pre or post aerosolization. It is likely that virus is less protected from inactivation in smaller particles, which may travel further, than in larger ones. We have amended the text to reflect this comment.

Line 510 – should be gelatine?

This has been changed

Line 520 – should be ‘influenced by many factors’

This has been changed